# Comment on: “Wnt Binding Affinity Prediction for Putative Frizzled-Type Cysteine-Rich Domains”

**DOI:** 10.3390/ijms20194884

**Published:** 2019-10-02

**Authors:** Orlando Musso

**Affiliations:** INSERM, Univ Rennes, INRA, Institut NuMeCAN (Nutrition Metabolisms and Cancer), F-35000 Rennes, France; orlando.musso@inserm.fr; Tel.: +33-2-2323-4565

**Keywords:** collagen XVIII, endostatin, Wnt, β-catenin, modelling, extracellular matrix

Dear Editor,

I read with great interest the article: *“Wnt Binding Affinity Prediction for Putative Frizzled-Type Cysteine-Rich Domains* [1]*”* published by Agostino et al. in *International Journal of Molecular Sciences* on August 26, 2019. The aim of the authors was to estimate the Wnt-binding affinities of proteins bearing putative frizzled-type cysteine-rich domains (FZD_CRDs). Binding of Wnts to cell surface frizzled receptors activates the Wnt/β-catenin signaling pathway involved in cell fate and oncogenesis. Wnt proteins are lipid-modified, which is important for Wnt secretion and binding to FZD_CRDs [2]. The authors presented two models predicting FZD_CRD-Wnt binding affinities; the essential difference is that one of them adopts an *a priori* approach, forcing protein lipidation to contribute favorably to the predicted binding energy [1]. 

I was surprised by the dramatic fall by approximately three orders of magnitude in the binding affinity of collagen type XVIII (COL18A1) for Wnt ligands when applying the model which *a priori* forces protein lipidation as a favorable influence. Although the predicted low binding affinities remain within the range of many physiologically relevant protein-protein interactions, the authors provide neither experimental evidence nor further modeling that explains the difference in COL18A1 FZD_CRD-Wnt binding affinities between both models. To reconcile their predicted low binding affinity and our previous experimental evidence that COL18A1 FZD_CRD binds Wnt3a and inhibits canonical Wnt/β-catenin signaling [3,4,5], the authors speculated that COL18A1’s strong affinity for heparin could potentiate the interaction of its FZD_CRD with Wnts. Their hypothesis, although attractive, is at odds with the current experimental evidence on the domain structure of the COL18A1 protein [5,6], COL18A1 C-terminal processing and binding to heparin [5,7], to Wnt and to other FZD_CRDs [3]:

**Modular structure of COL18A1**: The authors use COL18A1 and endostatin as synonyms. However, the structural relationships between full-length COL18A1 and its endostatin domain [7] are relevant to the interpretation of the authors’ results. COL18A1 belongs to the multiplexin family of collagens with multiple triple helices and globular interruptions. Three variants (130; 150 and 170 kD) containing specific N-terminal globular domains—thrombospondin1-like, FZD_CRD, and the Domain of Unknown Function (DUF)-959, result from tissue-specific promoter usage and alternative splicing [8,9]. Endostatin is a C-teminal 20 kD heparin-binding globular domain common to all variants, which is released from COL18A1 through proteolytic cleavage and is detected in normal human serum at high levels [7]. 

**COL18A1 binding to heparin and cell surface glycans**: The authors hypothesized that COL18A1 binding to heparin with high affinity could potentiate COL18A1 FZD_CRD-Wnt interaction, thus compensating for the predicted low COL18A1 FZD_CRD-Wnt binding affinity. Although we have shown that full-length COL18A1 can be purified from human tissues by heparin affinity chromatography [5]; binding of COL18A1 to cell surface heparan sulfate proteoglycans (HSPGs) is unlikely to play a major role in the interaction of its FZD_CRD domain with Wnt for several reasons. First, we and others have shown that COL18A1 molecules are themselves HSPGs [5,10]; the COL18A1 glycoprotein core is surrounded by a bulky “cotton-candy”-like mass of covalently attached, negatively-charged heparan sulfate side chains [5,10]. As Wnts bind heparan sulfates [2], overexpression of full-length COL18A1 indeed increases Wnt/β-catenin signaling [5], probably by promoting Wnt availability. However, this is unrelated to the FZD_CRD, because overexpression of a COL18A1 N-terminal domain carrying heparan sulfate side chains but not the FZD_CRD, amplifies Wnt3a-induced Wnt/β-catenin signaling [4]. Conversely, the FZD_CRD of COL18A1 binds Wnt3a independently of heparan sulfate side chains [3], and the cell surface localization of the N-terminal globular domain of COL18A1 is lost upon deletion of the FZD_CRD domain [5]. Second, only when the N-terminal COL18A1 FZD_CRD domain is excised from the parent molecule, which releases it from the bulky HSPG molecule, does the FZD_CRD domain inhibit Wnt/β-catenin signaling, suggesting that negative charges of both COL18A1 and cell surface HSPGs hinder FZD_CRD-Wnt interactions [5]. Third, it is unlikely that *N*-glycans and sialic acid, which are covalently attached to the DUF-959 domain adjacent to the FZD_CRD, contribute to Wnt binding because the FZD_CRD binds Wnt3a independently of DUF-959 [3,5].

**COL18A1 binding to Wnt and to other FZD_CRDs**: First, Wnt3a binds the COL18A1 FZD_CRD domain in cell-free medium [3]. Thus, neither cell surface heparan sulfates nor heparan sulfate side chains are required for the interaction to occur. Second, our modeling indicated 100% probability that the predicted 3D structure of COL18A1 FZD_CRD matches SFRP3 and FZD8_CRD crystal structures. In this model, a polypeptide that specifically inhibits the interaction of COL18A1 FZD_CRD with Wnt3a lies at its solvent-exposed surface and partially overlaps key interacting residues inferred from mutation analysis [5]. Third, other than binding Wnts at the cell surface, COL18A1 FZD_CRD forms homodimers and heterodimerizes with FZD1 and FZD8_CRDs, which could contribute to modulate Wnt/β-catenin signaling [3]. 

In conclusion, our previously published data show that COL18A1 FZD_CRD directly binds Wnt3a and other FZD_CRDs. As COL18A1 is itself an HSPG, the hypothetical role of heparin-binding in FZD_CRD-Wnt interaction may be hindered by negative charges from both COL18A1 and cell surface HSPGs. Moreover, our experimental data showed that the COL18A1 FZD_CRD domain interacts with Wnt and inhibits β-catenin signaling independently of heparin-binding sequences [3,4,5]. Mathematical modeling of interaction dynamics and binding affinities of FZD_CRD domains with putative partners may be enriched by experimentally modelling the challenging context of the cell surface matrix to generate digital data by quantitative dynamic approaches such as surface plasmon resonance.

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
