# Peer review of "Comment on: “Wnt Binding Affinity Prediction for Putative Frizzled-Type Cysteine-Rich Domains”"

_ijms, 2019, doi:10.3390/ijms20194884_

Round 1
Reviewer 1 Report
Current communication “Letter to the Editor: Wnt binding affinity prediction for putative frizzled-type cysteine-rich domains” by Orlando Musso comments on recently published article “Wnt binding affinity prediction for putative frizzled-type cysteine-rich domains” by Agostina and Pohl. This study which was basically to estimate the Wnt-binding affinities of proteins bearing FZD_CRD domains, presents two models to estimate binding affinities. One of the models assume binding of lipid to protein contributes favorably to estimated binding energies resulting in variation of binding affinity by three orders of magnitude to the other model. The low binding affinity model was justified by hypothesizing that COL18A1 binding to heparin will increase the likelihood of COL18A1-Wnt interactions. The published study was not supported by any new experimental evidence other than certain assumptions.
Author points out that although the published predictions are interesting, they are against available experimental evidence previously brought out by their team. Towards this, the author emphasizes the following established evidences which invalidates the published hypothesis (1) COL18A1 domain organization and existence of tissue specific variants (2) COL18A1 FZD_CRD interaction with Wnt3a & independent inhibition of Wnt/b-catenin signaling; COL18A1 HSPG lacking FZD_CRD domain can amplify Wnt-induced b-catenin signaling (3) COL18A1 FZD_CRD binds to Wnt3a regardless of heparin sulfate side-chains and (4) structural modeling of COL18A1 FZD_CRD matching SFRP3 and FZD8_CRD crystal structure, where polypeptide that inhibits COL18A1 FZD_CRD interaction with Wnt3a which was established by mutational analysis.
This article points out to the existing challenges and potential pitfalls in modeling binding affinities with presumptive assumptions. Additional research will certainly help in improving the modeling of binding affinities and can throw more light into these interactions. This article will definitely be of great interest to the research community.